# An Updated Review of Hypertrophic Scarring

**DOI:** 10.3390/cells12050678

**Published:** 2023-02-21

**Authors:** Manjula P. Mony, Kelly A. Harmon, Ryan Hess, Amir H. Dorafshar, Sasha H. Shafikhani

**Affiliations:** 1Department of Surgery, Division of Plastic & Reconstructive Surgery, Rush University Medical Center, Chicago, IL 60612, USA; 2Department of Medicine, Division of Hematology and Oncology and Cell Therapy, Rush University Medical Center, Chicago, IL 60612, USA; 3Cancer Center, Rush University Medical Center, Chicago, IL 60612, USA

**Keywords:** normal (acute) wound healing, hypertrophic scar, keloids, animal models, treatments

## Abstract

Hypertrophic scarring (HTS) is an aberrant form of wound healing that is associated with excessive deposition of extracellular matrix and connective tissue at the site of injury. In this review article, we provide an overview of normal (acute) wound healing phases (hemostasis, inflammation, proliferation, and remodeling). We next discuss the dysregulated and/or impaired mechanisms in wound healing phases that are associated with HTS development. We next discuss the animal models of HTS and their limitations, and review the current and emerging treatments of HTS.

## 1. Introduction

Wound healing is a complex physiologic process in which the body attempts to replace destroyed and damaged tissue with newly generated tissue and restore the skin’s barrier functions. It is an overlapping and sequential process of hemostasis, inflammation, proliferation, and remodeling that involves communication between many different cell types [1]. When this process does not occur in a sequential and finite manner, aberrant wound healing with hypertrophic scarring (HTS) or keloids may occur. These fibroproliferative disorders can be appreciated as elevated scars above the skin level with abundant deposition of extracellular matrix (ECM) components, especially collagen [2]. Although HTS and keloids are often used interchangeably, they are not the same. In HTS, excess scarring is limited to the original site of injury, whereas in keloids, scarring can extend beyond the original wound and is often regarded as a form of benign skin tumor [3,4].

Scarring is a major clinical problem, affecting some 100 million patients in the developed world alone [5]. The reported prevalence of hypertrophic scarring ranges from 32 to 72% [6,7]. Hypertrophic scars are particularly prevalent among adult burn patients, with those with darker skin, younger age, female sex, burns greater than 20% of total body surface area (TBSA), and burns on the neck and upper limbs experiencing the highest risk [6,8]. Following burn injury, nearly 75% of patients develop neuropathic pain [9]. Factors such as scar height, pigmentation, vascularity, and hyperplasia have been associated with increased levels of pain [9]. In one study, nearly 60% of patients who underwent bilateral reduction mammoplasty or median sternotomy incision developed HTS postoperatively, with an increased risk in those who were young [10]. Keloids have been reported in all ethnic groups, but they are significantly more common in individuals of African, Asian, and to a lesser degree, Hispanic descent, with the incidence ranging from 0.09% amongst the European white population, to 16% in the black population in Africa [11,12,13].

Severe HTS may result in scar contractures which can be significantly disfiguring and disabling and may lead to loss of mobility and affect patients’ ability to carry out routine daily activities [14,15]. In patients with severe burns, HTS is associated with decreased quality of life and delayed reintegration into society, in part due to the effect on self-esteem and the resultant desire to hide the scarring [16]. Globally, the wound care cost is estimated to be nearly $20.8 billion annually, with $4 billion per year associated with HTS treatment in the United States alone [17]. Hypertrophic wound care remains one of the largest markets without definitive drug therapy. The global hypertrophic and keloid scar treatment market size is expected to reach $37.9 billion US dollars by 2026 with a compound annual growth rate (CAGR) of 9.9% [18].

Hypertrophic scars typically occur in the second to third decade of life and present 1–2 months following injury [7]. The scar experiences a rapid growth phase for the first 6 months, followed by regression [7]. HTS arises as increased induration and dyspigmentation limited to the site of initial injury in areas of high tension, such as the shoulders, neck, prosternum, knees, and ankles [7]. Diagnosis of hypertrophic scarring is made clinically. Scoring systems such as the Vancouver Scar Scale, Seattle Scar Scale, Hamilton Scar Scale, and Patient and Observer Scar Assessment Scale may be used to assess the degree of hypertrophy [19]. These scales are based on clinical parameters such as lesion thickness, color, pliability, pain, and itching; however, the resulting scar scores are variable, as they are based on subjective clinical assessment. Combining the scar scales with more objective data such as high-resolution ultrasound scanning may be beneficial [20].

In this review article, we provide an overview of normal (also known as, acute) wound healing phases; namely, hemostasis, inflammation, proliferation, and remodeling. We next provide an updated review of the dysregulated and/or impaired mechanisms of HTS associated with each phase of wound healing. We then discuss the animal models of HTS and their limitations, and review the current and emerging treatments of HTS.

## 2. Overview of Normal Wound Healing

To gain better understanding of the pathophysiology underlying HTS, it is essential to appreciate the processes underlying normal (acute) wound healing in the acute setting. Normal wound healing occurs in four overlapping and complex phases; namely, hemostasis, inflammation, proliferation, and remodeling (Figure 1).

### 2.1. Phase 1: Hemostasis

Hemostasis begins immediately after injury and could last for several hours. As an immediate response to limit blood loss after injury, the blood vessels’ smooth muscle contracts via vasoconstrictors, such as endothelin, released by the damaged endothelial cells [21]. This is followed by blood clot formation, which occurs in two steps: primary hemostasis and secondary hemostasis. During primary hemostasis, rearrangement and transformation of the actin cytoskeleton occur in platelets, allowing a change in their morphology from disk-shaped to fried egg-shaped cells. This, in turn, causes platelets to interact with each other and the surrounding extracellular matrix (ECM) through activated integrins, allowing for the development of the platelet plug [21,22].

During secondary hemostasis, thrombin becomes activated via the intrinsic and extrinsic coagulation pathways [23]. Activated thrombin cleaves soluble fibrinogen into fibrin and cross-links them to form fibrin mesh, which is incorporated into the fibrin clot at the site of injury to form a thrombus, which enmeshes aggregated platelets and leukocytes into a stronger structure known as the platelet plug [22]. The platelet plug serves three important functions during wound healing: to prevent blood loss after injury, to serve as a source of chemokines and growth factors needed to initiate the inflammatory phase, and to function as a provisional scaffold for inflammatory leukocytes migration into damaged tissue [24,25].

### 2.2. Phase 2: Inflammation

Following injury, the inflammatory phase begins within minutes, peaks in 2–3 days, and can last 1–2 weeks, depending on the extent of the injury [1]. The primary functions of the inflammation phase during wound healing are to protect wounds against invading pathogens and to jumpstart the subsequent inflammatory and non-inflammatory responses needed for proper healing [26,27]. The inflammatory phase can be divided into early and late phases. During the early phase of inflammation, endothelial cells increase the expression of adhesion molecules, resulting in the recruitment and extravasation of inflammatory cells, such as neutrophils, monocyte, lymphocytes, and mast cells [28,29]. Leukocytes recruitment is mediated by pro-inflammatory cytokines, such as interleukin-1 (IL-1), IL-6, and tumor necrosis factor alpha (TNF-α), which are released from degranulating platelets, keratinocytes, endothelial cells, and tissue-resident macrophages [1,30,31]. Upon arrival into the wound, monocytes differentiate into pro-inflammatory M1 macrophages, which function to further amplify inflammatory responses by producing more pro-inflammatory cytokines, and assist neutrophils in destroying invading pathogens [32,33]. During the late phase of inflammation, the macrophages polarize into the anti-inflammatory M2 phenotype, which play pivotal roles in the resolution of the inflammatory responses and in the initiation of the proliferation phase through the production of a spectrum of angiogenic and growth factor mediators, such as vascular endothelial growth factor (VEGF), PDGF, and FGF2 [33,34,35].

### 2.3. Phase 3: Proliferation

The proliferative phase, also known as the new tissue regeneration phase, begins approximately 3 days after injury and lasts for about 2–3 weeks. The main events during the proliferative phase are provisional matrix replacement with granulation tissue, angiogenesis, and re-epithelization [36]. Initially, fibroblasts migrate to the site of injury in response to mediators released by platelets and macrophages, such as PDGF, transforming growth factor beta (TGF-β), and connective tissue growth factor (CTGF) [37]. To replace the provisional matrix with granulation tissue, fibroblasts release extracellular matrix (ECM) components (primarily type III collagen, fibronectin, proteoglycans, and hyaluronic acid) [38,39]. Granulation tissue is composed of ECM components, fibroblasts, proliferating blood vessels, macrophages, and lymphocytes, and it is an important indicator of wound healing progression [40].

During re-epithelization, M2 macrophages and keratinocytes produce and release EGF and TGF-β, which in turn induce proliferation and cell migration in epithelial cells bordering the wound edges to re-establish the epidermis integrity at the wound site [41]. Stem cells from hair follicles and sebaceous glands differentiate into keratinocytes to aid in this process [42].

Angiogenesis involves the creation of new vasculature that is 3 to 10 times denser than what is found in normal tissue [43]. It is critical in facilitating the transport of immune cells, oxygen, and nutrients for the cells participating in healing [43]. Angiogenesis is triggered by local hypoxia and several soluble factors, including VEGF (most prominent factor), PDGF, fibroblast growth factor-basic (bFGF), the serine protease thrombin, and members of the TGF-β family [44,45,46,47]. Following the completion of wound healing, most of the newly formed capillaries will regress [43].

### 2.4. Phase 4: Remodeling

The remodeling phase begins 2–3 weeks following injury and can last up to a year or even longer [48]. Matrix maturation and tissue remodeling depend on the balance between the degradation of extracellular matrix (ECM) components in granulation tissue and their replacement by connective tissue components, namely collagen I. Early in the remodeling phase, ECM components (e.g., collagen III, fibronectin, and hyaluronic acid) are degraded by matrix metalloproteinases (MMPs) [49]. Because of the destructive nature of the MMPs, they are tightly regulated by the tissue inhibitors of metalloproteinases (TIMPs) [50]. Moreover, fibroblasts differentiate into myofibroblasts which produce thick bundles of collagen I to replace most of the collagen III [51]. Over time, collagen I fiber bundles increase in diameter, resulting in increased wound tensile strength; however, the healed tissue never fully regains the properties of uninjured skin, resulting in a mostly acellular and avascular scar [50]. Scar tissue contains collagen bundles that are smaller and more disorganized and, thus, prone to dehiscence [52,53]. Over time, wound contraction occurs as the result of myofibroblasts bringing the wound edges together with the contractile function of their actin filaments [54].

## 3. Hypertrophic Scarring Associated with Wound Healing Phases 

A large body of evidence suggests that excessive inflammation generates pro-fibrotic molecules, which in turn activate fibroblasts, resulting in HTS [55]. In addition, excessive angiogenesis and prolonged re-epithelialization can extend the release of pro-fibrotic growth factors [56,57]. In the last few years, many biomolecules have been implicated in HTS; however, their exact mechanisms have yet to be fully elucidated, in part due to the complexity and overlapping nature of wound-healing processes. Here, we will examine each phase of wound healing with respect to HTS formation.

### 3.1. Phase 1: Hemostasis

The fibrin provisional matrix deposited during hemostasis has been implicated in the activation of myofibroblasts and the formation of HTS [58]. In fact, high-density fibrin clot deposition during the early phase of healing may predict the formation of HTS (Figure 2), as calculated by a multiscale mathematical model [59]; however, more studies related to fibrin content and rate of fibrinolysis in experimental models are required to validate the role of the fibrin provisional matrix in the formation of HTS. In addition, during hemostasis, platelets release a multitude of pro-fibrotic growth factors such as PDGF, VEGF, TGF-β1, and CTGF, which have been linked to the formation of HTS (Figure 3) [60]. Interestingly, platelet-rich plasma (PRP) obtained from platelets of the peripheral blood is considered to be a therapeutic option for HTS, as it reduces the expression of pro-fibrotic molecules such as TGF-β1 and CTGF [61]. These reports suggest that while naïve platelets may be anti-fibrotic in nature when activated excessively, they can contribute to HTS development. Clearly, more studies are needed to evaluate the role of naïve versus activated platelets in HTS.

### 3.2. Phase 2: Inflammation

Excessive inflammation (Figure 2) is the best elucidated pathophysiological reason for HTS formation [55]. As such, many of the accepted therapeutics target inflammation [55]. Excessive infection and tissue necrosis in severe burn wounds cause increased pathogen-associated molecular patterns (PAMPs) and damage-associated molecular patterns (DAMPs), toll-like receptor (TLR) signaling, and infiltration of inflammatory cells to the wound site [62,63,64]. 

Surprisingly, studies of HTS have found chemokine expression to be variable. In a study using the rabbit ear as a model for HTS, the expression of chemokines such as Chemokine (C-C motif) ligand 3 (CCL3), CCL7, and CCL13 maintained increased expression for 21 to 35 days, while CCL2, CCL4, CCL5, and chemokine (C-X3-C motif) ligand 1 (CX3CL1) were maintained at high levels for 21 to 56 days [65]. Another study reported that the expression of CCL3, CXCL1, CXCL2, CXCR2, C3, and Interleukin 10 (IL-10) was reduced in human HTS, 52 weeks following surgery [66]. In another study, SDF1/CXCR4 signaling was found to be increased in human HTS tissue [67]. The underlying reasons for this variability remain unknown and require future investigation.

Inflammatory cells release various factors such as interleukins, interferon, and growth factors [68]. Increased expression of pro-inflammatory and pro-fibrotic growth factors activate fibroblasts and are thus implicated in HTS formation [69]. Interestingly, in a study of HTS tissue at 3 h following surgery, the expression of certain pro-inflammatory factors such as IL-6, IL-8, and CCL2 was found to be reduced during the early phase of healing [70]. Intriguingly, inadequate pro-inflammatory responses have also been reported in hypofibrotic diabetic wounds early after injury, rendering them vulnerable to infection and impaired healing [71,72,73,74]. This delay in inflammatory responses during the acute phase of healing early after injury and its role in the formation of HTS should be further investigated. 

IL-6 is a major cytokine that influences the middle and late phases of healing, as it is involved in shifting inflammation from acute to chronic by enhancing monocyte recruitment, M2 macrophage polarization, and ECM deposition [75,76,77]. IL-6 is highly expressed in HTS and is considered to be a therapeutic target for the treatment of HTS [69,78]. The IL-6/STAT3 (Signal transducer and activator of transcription 3) pathway activates many of the genes required for ECM production and fibroblast proliferation, leading to HTS [79]. Other than the IL-6 and inflammatory chemokines, other inflammatory cytokines that are highly expressed in HTS include IL-1β, IL-4, IL-8, IL-17, IL-13, and IL-22 (Figure 3) [69,80,81]. Some of these cytokines, such as IL-4 and IL-13, have been under investigation as therapeutic targets for HTS [82]. The expression of IL-10, an anti-inflammatory cytokine and promising therapeutic molecule, has been found to be low in patients with hypertrophic scarring compared to those with non-hypertrophic scarring [66]. Some studies have suggested that IL-10 directly influences fibroblasts by activating the STAT3 or AKT signaling pathways [83]. It has also been reported that IL-10 reduces scar formation by regulating the TLR4/NF-kB pathway in dermal fibroblasts [84]. However, further investigation is required to elucidate the role of IL-10 in preventing HTS development. Additionally, the expression of other cytokines, such as IL-24, IL-36, IL-37, IL-1RA, and TNF-α, has been found to be low in HTS (Figure 3) [69]. TNF-stimulated gene 6 (TSG-6) has been found to suppress scarring by downregulating the IRE1α/TRAF2/NF-κB signaling pathway [85]. Moreover, alteration in the fatty acid metabolism influences inflammation and can result in excessive scarring [86,87]. In a recent study, the expression of sterol regulatory element-binding protein-1 (SREBP1) and fatty acid synthase (FASN) was shown to be reduced at mRNA and protein levels in pathological HTS and in HTS-derived fibroblasts [86]. In another study, the expression of fatty acid desaturase 1 and 2 (FAD1 and FAD2)—key enzymes in the polyunsaturated fatty acids (PUFAs) metabolism with demonstrated anti-inflammatory function [88]—were lower in keloids and keloid-derived fibroblasts [87]. However, the mechanism of altered lipid profile in HTS has not been explored. It is possible that alterations in lipid metabolism might influence HTS through changes in the inflammatory pathways, given that fatty acids play an important role in regulating inflammation [89,90]. 

### 3.3. Phase 3: Proliferation

Events of the proliferative phase, such as angiogenesis and ECM deposition, are highly active in HTS, whereas re-epithelialization is prolonged in HTS as keratinocytes remain continually activated (Figure 2) [91,92,93]. Consequently, the granulation tissue becomes denser during HTS formation than in normal scarring (Figure 2). In HTS, cells such as keratinocytes, endothelial cells, and fibroblasts release many pro-fibrotic growth factors such as TGF-β, PDGF, VEGF, and CTGF [94]. This pro-fibrotic environment, in turn, induces fibroblasts to produce more ECM proteins such as collagen, fibronectin, laminin, periostin, fibrillin, and tenascin; however, the expression of certain ECM proteins such as hyaluronic acid, dermatopontin, and decorin are found to be altered or reduced (Figure 3) [95]. Fibroblasts of the deep dermis are responsible for the production of additional factors such as osteopontin, angiotensin-II, and peroxisome proliferator-activated receptor (PPAR)-α and contribute to scarring more than fibroblasts of the superficial dermis [96]. Recently, it has been revealed that fibroblasts in the upper dermis also contribute to scarring by producing IL-11, which in turn activates myofibroblasts [97]. TGF-β plays an important role in the formation of HTS, and the TGF-β/SMAD (Suppressor of Mothers against Decapentaplegic) signaling pathway is considered to be a potential therapeutic target of HTS [98]. Molecules such as SMAD interacting protein and bacterial PAMPs such as lipopolysaccharides (LPS) may induce HTS by enhancing the TGF-β1/SMAD signaling pathway [99,100]. 

Endothelial cells isolated from porcine burn wounds show that endothelial dysfunction and altered expression of angiogenic genes such as endothelin-1, angiopoietin-1, angiopoietin-2, and angiogenin may result in HTS (Figure 3) [57,101]. In turn, angiogenesis is stimulated by the microvesicles released from the myofibroblast [102]. Factors released from these vesicles may result in HTS, as many of them are pro-fibrotic in nature [102,103]. During hypertrophic scarring, keratinocytes remain in their activated state for a prolonged duration of time [92]. Dysregulation in the Notch signaling of keratinocytes may also contribute to HTS formation [104]. Notch 1 signaling and intracellular domains such as Jagged1 and Hes1 are highly expressed in the epidermis of hypertrophic scar patients [104]. This leads to the enhanced expression of pro-fibrotic factors, such as TGF β1, TGF β2, CTGF, IGF-1, VEGF, and EGF (Figure 3) [104]. In addition, epithelial–mesenchymal transition increases ECM deposition and has been shown to contribute to HTS formation [105]. Moreover, keratinocytes produce HMGB1, which activates fibroblasts, resulting in HTS formation [106]. However, certain factors released from keratinocyte-like pigment epithelium-derived growth factor (PEDF) are associated with reduced angiogenesis and HTS formation (Figure 3) [107]. Interestingly, among different growth factors, FGF-2 has an anti-scarring effect since it up-regulates the expression of MMP-1 and hepatocyte growth factor (HGF), although further investigations are required to clarify its therapeutic potential [108,109]. 

### 3.4. Phase 4: Remodeling

In HTS, the balance of ECM synthesis and remodeling is dysregulated [110]. Both fibroblasts and myofibroblasts continue to deposit collagen III and collagen I in HTS [111]. The persistence of myofibroblasts due to defects in apoptosis results in the deposition of excessive fibrous collagen I and scarring (Figure 2) [112,113,114]. The presence of nodules containing myofibroblasts is a peculiar feature of HTS [50]. Mechanical stretch and TGF-β can stimulate the differentiation of fibroblasts to myofibroblasts, contributing to HTS formation [115,116].

Metalloproteinases (MMPs), such as MMP1 and MMP7, are downregulated during HTS formation, resulting in reduced degradation of ECM components such as collagen I, collagen III, and fibronectin ([117,118] and Figure 2). Administration of MMP1 has been shown to improve scarring [119]. The tissue inhibitors of metalloproteinases (TIMPs), such as TIMP1 and TIMP2, reduce the action of MMPs during HTS development (Figure 3) [110]. In contrast, expression of MMP2, MMP9, and MMP13 are shown to be increased in HTS (Figure 3) [110,118]. This upregulation may be a compensatory response to elevated levels of ECM in HTS, but it remains unclear and requires future investigation. In HTS, reduced expression of matrix remodeling proteins results in the disorganization of ECM components [50,113]. Treatment with decorin, a matricellular protein involved in collagen fiber organization, has been shown to reduce HTS formation [120]. In addition, targeting the lysil hydroxylase enzyme, involved in the formation of pyridinoline cross-links, reduces the activity of fibroblast proliferation by regulating TGF-β1 [121].

## 4. Animal Models of Hypertrophic Scarring

While two-dimensional and three-dimensional cell culture-based in vitro models can be useful for investigating the mechanism of fibroblast in producing excessive ECM and potential therapeutic molecules, the absence of immune and vascular components in these models limits the physiological relevance of the findings emerging from these studies with respect to the mechanisms underlying HTS formation in tissue [122,123]. In the past several decades, many attempts have been made to develop animal models of HTS in different species. Despite these attempts, there are no animal models that can fully recapitulate HTS in humans. Descriptions of each animal model for HTS, as well as their advantages and disadvantages, are summarized in Table 1. 

The rabbit ear model has been widely used to study HTS formation despite the involvement of chondrocytes during the healing, where skin and perichondrial layers are removed from the ventral side of the rabbit ear to generate an HTS-like condition [124,125]. The advantages of this model include the simplicity of the procedure, ease of handling of the animal, and ability to create multiple wounds; however, the ventral side of the ear is difficult to handle because of its low thickness, and precaution needs to be taken to avoid damage of the underlying cartilage during the procedure [124,125]. To reduce damage to the rabbit ear cartilage during the procedure, cryosurgery has been attempted to remove the perichondrial layer [126]. In another rabbit ear model for HTS, thermal burn injury has been attempted to create a more elevated scar within a shorter duration which better mimics an HTS condition in humans [127]. However, thermal injury has to be precisely controlled to avoid variability in scarring [127]. Injecting anhydrous alcohol into the subcutaneous and superficial fascia regions of the dorsal skin of a rabbit has been used to model HTS [128]; however, this model appears to be more appropriate for skin fibrosis than the HTS due to the absence of a healing response. 

Deep burn injury to the dorsal side of porcine skin creates raised scar tissue and has been used in some studies as a model for HTS [57]. Although there are structural similarities between human and pig skin, the high costs associated with the production of this animal model and the difficulty in handling it have lessened its popularity for HTS studies. 

Several groups have also attempted to develop rodent (mouse and rat) models for HTS [129,130,131]. These murine models are inexpensive to produce and easy to handle, but wound healing patterns in rodents differ from that of humans due to the rapid contraction of the panniculus carnosus muscles [132]. To mitigate the effect of rapid wound contraction in rodents, splinting excision wounds have been attempted [130]. For example, splinted full-thickness skin wounds in rodents recapitulate mechanical tension in the wound bed, and the lack of neo-epithelium in this model amplifies myofibroblast function, culminating in hypertrophic features, which are similar to HTS in humans [131]. Similarly, mechanical pressure applied to a wound by a biomechanical loading device also produces HTS-like features in mice [133]. 

C–X–C motif chemokine receptor 3 (CXCR3)-deficient mice develop thick keratinized scars and have been used in some studies to model HTS, but deficient dermal maturation with poor collagen content has been observed [134,135,136]. Hence, the role of CXCR-3 and its effect on matrix development require further investigation. 

By resecting the abdominal wall muscle on the ventral side of mice that produces contractile forces, another murine wound model for scarring has been created, but it is not comparable with the healing mechanism underlying HTS [137].

Some attempts have been made to develop a xenograft model of HTS by grafting tissue from human HTS onto nude mice [138,139]. These mice displayed scar thickness and collagen bundle orientation and morphology resembling human HTS [129]. However, a lack of an immune response and difficulty in maintaining nude mice may obstruct the study of therapeutic molecules in this model. 

Developing an ideal animal model for HTS is exceptionally challenging, as the scar endotype is difficult to control in experimental settings [140]. The aforementioned animal models all fall short; therefore, developing an ideal animal model is essential to support studies related to the formation of and therapy for HTS. 

## 5. Conventional and Emerging Treatments for Hypertrophic Scarring

### 5.1. Conventional Therapies

Treatments of hypertrophic scars often focus on correction of factors that are associated with pathological scar development as described above. These include wound stabilization, minimizing mechanical irritation, balancing wound healing phases, attenuating pro-fibrotic mechanisms, inducing anti-fibrotic mechanisms, and promoting the remodeling of collagenous scar components. Published guidelines on the treatment of hypertrophic scars and keloids include many different modalities without one single, widely accepted protocol [3,141]. Several treatments and techniques have been shown to prevent the development of hypertrophic scar development. (These conventional treatments have been summarized in Table 2). Reduction in tension on the dermal layer when closing wounds is effective and can be achieved with fascial and subcutaneous tensile reduction sutures in wounds of adequate depth [142]. Additionally, dermal closure using sutures arranged in a zig-zag pattern or using z-plasties should be performed whenever possible [142,143]. Closure with 3–0 VLoc 90 barbed suture (VLoc, Covidien, North Haven, CT, USA) compared to interrupted suture with 4–0 nylon produced significant improvements in the Vancouver scar scale (VSS) and patient and observer scar assessment scale (POSAS) scores in patients undergoing anterolateral thigh flap procedures with identical methods of deep closure between groups [144]. 

Following the closure of initial wounds, several therapies can also be applied early in the healing process. Similarly to the aforementioned suturing techniques, wound stabilization using paper tape or silicone sheets can also prevent the dermal inflammation that contributes to hypertrophic scar and keloid formation [145]. Wound compression using pressure garment therapy at 15–40 mmHg has been shown to improve outcomes [146]. Regarding ideal pressure, one review of pressure garment therapy for the treatment of burn wounds found that the application of pressure at 17–24 mmHg resulted in improved scar height, softness, and cosmetic appearance compared to a pressure below 5 mmHg [147]. Cohesive silicone sheets that added pressure to the wound also outperformed silicone gel sheets in improving scar assessment scale scores [148]. Intermittent application of pressure through regular massage therapy has not been shown to improve outcomes, suggesting that constant pressure must be applied [149]. 

Topical agents applied to heal wounds have also been shown to reduce hypertrophic scar formation, including flavonoids and silicone cream [150,151]. The local injection of Botulinum toxin-A postoperatively has also been shown to significantly improve scar assessment scale scores compared to controls [152,153,154]. In a recent study of optimal dosing of Botulinum toxin-A, postoperative injections of 8 units showed significantly improved Stony Brook Scar Evaluation Scale (SBSES) scores compared to the injections of 4 units [155]. The culture of human fibroblasts with Botulinum toxin-A resulted in decreased proliferation, migration, and secretion of pro-fibrotic factors, while JNK phosphorylation levels were increased, providing evidence for possible mechanisms of this benefit [156]. 

Scar revision is the simplest method of treating pre-existing HTS and encompasses procedures aimed at excisional debulking of hypertrophic scar tissue ([157] and Table 2). Closure during these procedures is specifically directed at providing favorable cosmetic results and should employ the methods described above for prophylaxis against scar recurrence. To be effective, scar revisions should be performed over 1 year from the original injury to give adequate time for the scar to mature [150], as immature scars are prone to hypertrophic healing and give poor results after scar revision [158].

However, excision may not be necessary, as more conservative measures have proven to be effective. For example, in one study, mechanical disruption of existing hypertrophic scars using microneedle roller therapy improved scar pigmentation to resemble surrounding tissue more closely, and significantly improved both the mean patient satisfaction scale (PSS) and observer satisfaction scale (OSS) between preoperative and postoperative sampling [159]. Another study found that microneedle therapy improved modified Vancouver scar scale (mVSS) scores significantly more than carbon dioxide (CO_2_) laser therapy for hypertrophic scars [160]. This benefit may be explained by microneedle therapy disrupting existing collagen and stimulating the release of MMP-9 [161]. 

Pharmacologic agents have also been used frequently in the treatment of hypertrophic scars, with common agents including corticosteroids, chemotherapeutic agents, and Botulinum toxin-A. Corticosteroids provide benefits through their potent anti-inflammatory effects and are believed to induce local vasoconstriction when applied to hypertrophic scars and keloids. Tapes and plasters containing corticosteroids effectively treat hypertrophic scars and keloids when applied to these lesions and should be positioned to avoid contact with surrounding tissue [145]. The most common use of corticosteroids in the treatment of hypertrophic scars and keloids by far is the intralesional injection of triamcinolone (TAC). A recent literature review and meta-analysis of this therapy found that compared to 5-FU and verapamil, TAC alone improved scar vascularity [162]. However, TAC therapy also had higher rates of skin atrophy and telangiectasias, especially at the commonly used dose of 40 mg/mL [162]. Significant differences in favor of other agents were found for scar height (5-FU, TAC + 5-FU), scar pliability (TAC + 5-FU, Botulinum toxin-A), scar pigmentation (TAC + 5-FU), VSS score (TAC + 5-FU, TAC + platelet rich plasma), and POSAS score (bleomycin) when compared against TAC alone [163]. A study of TAC vs. TAC + 5-FU found significant differences favoring TAC + 5-FU in mean reduction in scar height, overall POSAS score, and the overall rate of efficacy. Rates of telangiectasias (commonly known as “spider veins”), skin atrophy, hypopigmentation, and recurrence were significantly higher in the group receiving TAC, while the rates of ulceration were significantly higher in the group receiving TAC + 5-FU [164]. A literature review and meta-analysis of intralesional Botulinum toxin-A injection found significantly improved visual analog scale (VAS) scores compared to intralesional corticosteroid and placebo injection [165]. In a split-scar study of patients with existing hypertrophic scars, injection of Botulinum toxin-A was found to significantly improve mean VSS score pre- and post-treatment as compared to the placebo control [166]. 

The energy-based therapy is well established as a treatment modality for hypertrophic scars and keloids, with its use dating back to the 1980s [167]. Lasers are the mainstay of energy-based treatments, with a multitude of different laser devices utilizing different wavelengths for specific targets [168]. Laser therapy is often used in the treatment of formed hypertrophic scars but can also be used preventatively in the early postoperative period. In a split-scar study of patients undergoing total knee arthroplasties, scar treatment with a 595 nm pulsed-dye laser was associated with significantly improved overall VSS scores compared to an untreated scar [169,170,171,172,173,174]. 

The guidelines for the use of energy-based treatment for acne scars have included specific recommendations for use with hypertrophic acne scars and keloids. In patients with active acne, a 1064 nm ND:YAG laser is preferred, and pulsed-dye vascular lasers are the laser treatment of choice for hypertrophic acne scars. Pulsed-dye lasers (PDL) may also be used to assist with the delivery of 5-FU and/or TAC. Non-laser devices, including Tixel (Novoxel, Ltd., Berlin, Germany) and EnerJet (PerfAction Technologies Ltd., Rehovot, Israel), were also recommended for the treatment of hypertrophic acne scars [175]. Similar guidelines for traumatic scars recommend non-ablative fractional laser (NAFL) for hypertrophic scars, except in the presence of significant thickness and textural irregularity, where ablative fractional laser (AFL) therapy is preferred [169]. In a study comparing no laser treatment, CO_2_ laser treatment alone, and intense pulsed light (IPL) + CO_2_ laser, both treatment groups had statistically significant improvements in POSAS score and Manchester scar scale (MSS) score compared to the placebo, without significant difference between the treatment groups. The only significant difference between treatment groups was in favor of the combination therapy for scar color and texture, indicating that CO_2_ alone is sufficient and IPL can be used for an additional benefit for these specific factors [176]. Regarding protocols for CO_2_ laser, a study of varying densities for fractional CO_2_ laser treatment found that high (25.6%) density significantly improved VAS and POSAS scores compared to low (7.4%) and medium (12.6%) densities in treating mature hypertrophic burn scars [170]. A split-scar study of low-energy CO_2_ fractional laser treatment showed significantly improved POSAS scores for all elements except for patient-scored irregularity compared to the control for pediatric patients with early-stage hypertrophic burn scars [171]. A study of CO_2_, PDL, and CO_2_ + PDL for the treatment of hypertrophic burn scars found significant improvements in posttreatment POSAS for all treatment groups. Focused analyses found that scar height was improved by PDL or CO_2_ + PDL for scars <0.3 cm, and a significant reduction in scar height was achieved by CO_2_ + PDL only for scars older than 9 months. Although the guidelines for hypertrophic acne scars include the use of laser-assisted delivery of corticosteroids, a study of fractional ER:YAG laser alone or in combination with topical clobetasol found no significant benefit from the addition of steroids, with both treatment groups achieving significant posttreatment improvements in scar thickness and POSAS scores [172,175]. Recently, studies have compared IPL to non-laser therapies. Significant differences in scar pliability, hyperpigmentation, and median VAS favored IPL vs. silicone sheet, but significant differences in VAS and histopathological characteristics favored cryotherapy vs. IPL [173,174]. 

### 5.2. Emerging Treatments

Given the prevalence of hypertrophic scarring, new treatments are continually developed. Intralesional TAC, for example, was found to improve scar height, pliability, and pigmentation when combined with 5-FU and reduced the number of treatment sessions and remission time when combined with 1550 nm erbium glass fractional laser treatment (Table 3) [163,164,177,178]. While Botox A with TAC showed no difference in scar appearance, it significantly reduced pain and pruritis [179]. Scars treated with RFA plus verapamil and 5-FU experienced the fastest scar volume reduction with relief of symptoms and hyperemia compared to either agent alone [180]. Additionally, the combination of intense pulse light (IPL) and CO_2_ laser significantly improved scar color and texture [176]. The combination of lasers with 5-FU and/or TAC delivered intralesionally or via laser assistance has thus been recommended for the treatment of hypertrophic acne scars [169,175]. 

The role of angiotensin II in scar activity has recently been examined [181]. Human dermal fibroblasts treated with losartan, an angiotensin II type 1 receptor antagonist, displayed decreased contractile activity, fibroblast migration, gene expression of TGF-β1, type 1 collagen, and MCP-1, while reducing monocyte migration and adhesion [181]. In rat models, the consumption of losartan showed decreased cross-sectional area and elevation index in scars, with decreased α-SMA+ and CD68+ during immunostaining [181]. Another in vivo model demonstrated a reduced incidence of hypertrophic scarring with decreased inflammation, collagen and fibroblast cellularity, vascularization, and myofibroblast activity with the topical administration of oxandrolone and hyaluronic acid gel [182]. Clinically, the administration of dipeptidyl peptidase-4 inhibitors was shown to reduce the risk of hypertrophic scarring and keloid onset by less than half in patients who underwent sternotomy, while 1,4-diaminobutane (1,4 DAB) in breast reduction patients resulted in significantly greater scar satisfaction and less scar hardness measured by Rex Durometer [183,184]. 

Autologous fat grafting also presents as a novel therapy to improve the function and appearance of scars. While the underlying mechanism is unknown, exposure to adipocytes decreased the expression of the myofibroblast marker α-SMA and ECM components [185]. The reprogramming of myofibroblasts was found to be triggered by BMP-4 (bone morphogenetic protein 4) and activation of PPARγ (peroxisome proliferator-activated receptor gamma) signaling, which initiated tissue remodeling [185]. 

As is the case in many other fields of medicine, stem cells are also a promising therapeutic target for HTS. Mesenchymal stem cells (MSC) isolated from the mouse whisker hair follicle outer root sheath were applied to an in vivo full-thickness wound model [186]. A quantitative evaluation revealed reduced inflammation, cellularity, and collagen filaments, as well as thinner dermal and epidermal layers in the MSC-treated wounds, indicating a reduction in hypertrophic scars. Another study examined the effect of combined treatment with a non-ablative laser and human stem cell-conditioned medium on burn-induced hypertrophic scar formation [187]. The treatment group was found to have reduced erythema, trans-epidermal water loss, and scar thickness. 

Platelet-rich plasma (PRP) has also been identified as a promising therapy for scarring. In one study, primary dermal fibroblasts isolated from hypertrophic scars were cultured in a medium supplemented with 5% PRP or platelet-poor plasma (PPP) [61]. The PRP group was found to have reduced expression of TGF-β1 and connective tissue growth factor (CTGF) mRNA. Other studies have examined combination treatments with both PRP and ablative fractional CO_2_ lasers and have found the combination to be more beneficial than either treatment alone [188,189]. 

In addition, identifying the molecular targets for potential treatments is an ongoing source of investigation. Co-cultures of anti-inflammatory cluster of differentiation 206 (CD206)+ macrophages and fibroblasts showed decreased expression of fibrotic factors, such as type 1 and 2 collagen, alpha-smooth muscle actin, connective tissue growth factor, and TGF-β, with upregulation of MMP-1. IL-6 was also found to be increased in the medium, with an increase in anti-fibrotic gene expression when IL-6 was added to fibroblasts. Cytotherapy with cultured CD206+ macrophages or a direct administration of recombinant human IL-6 has been shown to dampen the expression of pro-fibrotic mediators (e.g., COL1A1 *, COL2A1 *, α-SMA *, CTGF *, and TGF-β1) in fibroblast in cell culture studies [190]. 

In vitro studies of fibroblasts have revealed that IFN-γ inhibits collagen synthesis [191]. IFN-γ knockout mice were found to have reduced wound closure, lower wound breaking strength, and dampened expression of collagen type 1A (COL1A1) and collagen type 3 A1 (COL3A1) mRNA, but a greater expression of MMP-2 (gelatinase) mRNA [191]. The study concluded IFN-γ may be involved in both the proliferation and maturation stages of wound healing and, therefore, may be a target for potential treatments. 

## 6. Conclusions

As this review illustrates, there has been significant knowledge gained in the field of hypertrophic scarring. A pro-fibrotic environment results in excessive collagen deposition and, therefore, hypertrophic scar formation. In this review article, and for the first time, we highlighted the defective and impaired mechanisms underlying HTS that are associated with each phase of wound healing (hemostasis, inflammation, proliferation, and remodeling). This was an attempt to demonstrate the multifaceted nature of the phase-specific dysregulations and impaired mechanisms that underlie HTS development. We further discussed the current animal models and their limitations in order to highlight the need for better animal models that can more closely reproduce the human condition with respect to HTS development. We also reviewed the current and emerging therapies, which further demonstrate the inadequacy of therapies to address HTS. There is still much to be discovered in regard to the underlying mechanisms contributing to HTS development. A better understanding of the impaired mechanisms underlying HTS would surely lead to the development of more effective targeted therapies to treat this debilitating and costly pathological condition.

## Figures and Tables

**Figure 1 cells-12-00678-f001:**
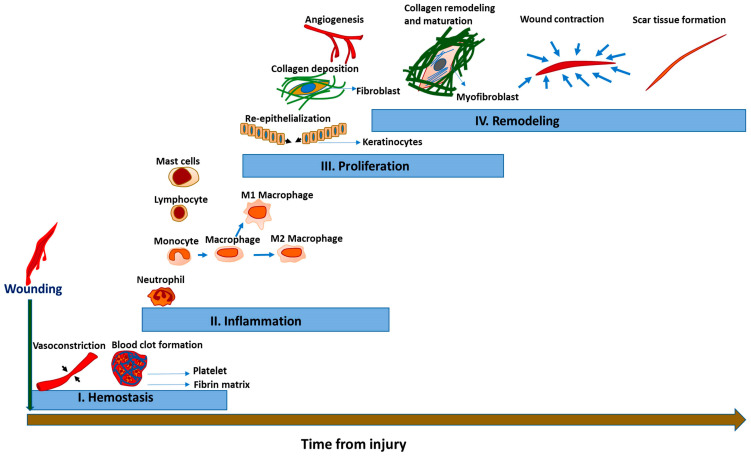
The phases of acute wound healing, including hemostasis (I), inflammation (II), proliferation (III), and remodeling (IV). Hemostasis begins soon after wounding with vasoconstriction and blood clot formation. This is followed by the infiltration of inflammatory cells. Then, re-epithelialization occurs with collagen deposition and angiogenesis during the proliferation phase. Finally, the remodeling phase occurs with collagen remodeling and maturation, wound contraction, and scar tissue formation.

**Figure 2 cells-12-00678-f002:**
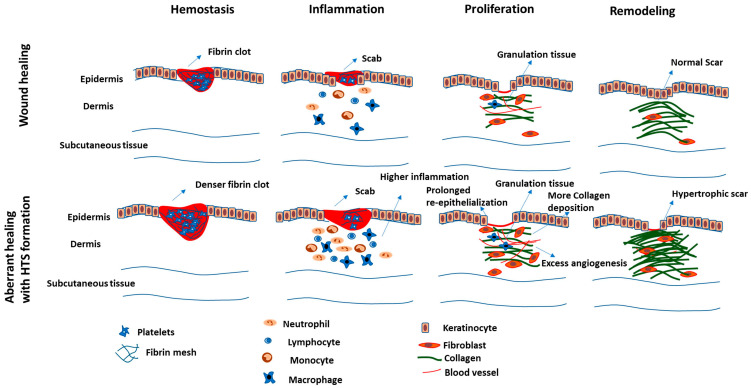
Phases of normal wound healing versus aberrant wound healing with the formation of hypertrophic scars. Events such as higher fibrin clot deposition, infiltration of higher number of inflammatory cells, prolonged re-epithelialization, and excess angiogenesis can result in excessive and improper collagen deposition and, therefore, formation of hypertrophic scars.

**Figure 3 cells-12-00678-f003:**
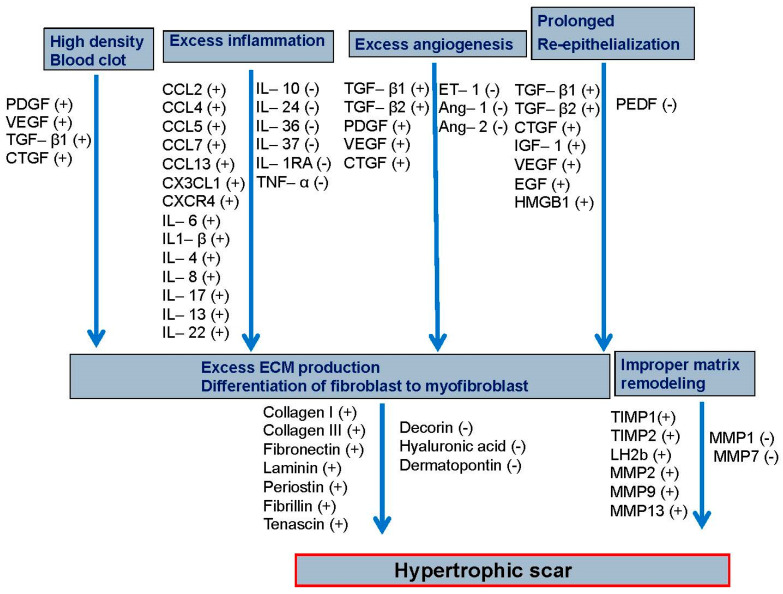
The events and molecules associated with hypertrophic scar (HTS) formation. The pro-fibrotic molecules generated from the high-density blood clot induce excessive inflammation, angiogenesis, and prolonged re-epithelization. The resultant excessive production of extracellular matrix and fibroblast differentiation and improper matrix remodeling then causes formation of hypertrophic scarring. Molecules with increased expression are denoted with (+), whereas those with decreased expression are denoted with (−). The players included in this figure are discussed in the text.

**Table 1 cells-12-00678-t001:** Animal models for hypertrophic scarring (HTS).

Model	Scar Location	Advantages	Disadvantages
Rabbit ear HTS * model [124,125]	Ventral side of rabbit ear	Simple, reliable model Ease of handling Possibility of creating multiple wounds	Skin of the ventral side is too difficult to handle because of low thickness Involvement of cells other than skin cells during healing, such as chondrocytes Risk of damaging the underlying cartilage
Modified rabbit ear HTS model—use of cryosurgery [126]	Ventral side of rabbit ear	Low risk of damaging the cartilage	Skin of the ventral side is too difficult to handle because of low thickness Involvement of cells other than skin cells during healing, such as chondrocyte
Modified Rabbit ear HTS model—application of thermal injury [127]	Ventral side of rabbit ear	Elevated scar within short duration compared to the typical rabbit ear HTS model	Skin of the ventral side is too difficult to handle because of low thickness Involvement of cells other than skin cells during healing, such as chondrocytes Uncontrolled thermal injury can cause variability in scarring effect
HTS model on rabbit by injecting anhydrous alcohol [128]	Dorsal skin	HTS-like appearance comparable to the rabbit ear HTS model Low cost Ease of handling	Absence of healing response
Burn hypertrophic model on porcine skin [57]	Dorsal Skin	Elevated scar comparable to human scar	High cost Difficult to handle
HTS model by splinting of rat wound [131]	Dorsal skin	HTS-like features by reducing the formation of neo-epithelium Low cost Ease of handling	Splinting may create a higher and more persistent tensional state
Scar on CXCR3 * deficient mouse [134]	Dorsal skin	Simple, reliable model Ease of handling	The model requires further validation
HTS model produced by grafting human xenografts on nude mice [139]	Dorsal skin	Establishment of human scar on an animal model	Difficulty in maintaining nude mice Absence of immune response in mice
HTS model by resecting abdominal wall muscle on mice [137]	Ventral skin, abdominal region	Simple and reliable method Ease of handling	Not comparable with general scar development after burn injury or trauma

* Abbreviations: HTS (hypertrophic scar); CXCR3 (C–X–C motif chemokine receptor 3).

**Table 2 cells-12-00678-t002:** Conventional treatments for hypertrophic scarring.

Treatment	Mechanism
Tensile reduction suture closure [142]	Reduces tension on the dermal layer when closing wound
Paper tape, silicone sheets [145,148]	Hydration, increased temperature, prevent dermal inflammation
Wound compression [146]	Reduces capillary perfusion, accelerated collagen maturation
Laser [169,170,171,172,173,174]	Destroys microvascularization, resulting in hypoperfusion and hypoxia
Silicone cream [150,151]	Hydration of the stratum corneum and cytokine-mediated signaling from keratinocytes to dermal fibroblasts
Flavonoids [150]	Anti-inflammatory, antioxidant, anti-bacterial
Botulinum toxin-A [152,153,154,155,156]	Decreases proliferation, migration, and secretion of pro-fibrotic factors from fibroblasts
Scar excision [157]	Removal of affected tissue
Microneedle [159,160,161]	Disruption of existing collagen, stimulation of MMP-9 * release
Corticosteroids [145,163]	Anti-inflammatory, local vasoconstriction
Botulinum toxin-A [163,165,166]	Decreases proliferation, migration, and secretion of pro-fibrotic factors from fibroblasts

* Abbreviations: MMP-9 (metalloproteinase 9).

**Table 3 cells-12-00678-t003:** Emerging therapeutics for hypertrophic scarring.

Treatment	Proposed Mechanism
Corticosteroids + 5-Fluorouracil [163,164,177,178]	Anti-inflammatory, local vasoconstriction, inhibit fibroblasts proliferation, decrease collagen synthesis
Laser + Verapamil + 5-Fluorouracil or corticosteroids [180]	Destroy microvascularization resulting in hypoperfusion and hypoxia, inhibit fibroblast proliferation, decrease collagen synthesis, anti-inflammatory
CO_2_ * Laser + Intense Pulse Light [176]	Destroy microvascularization resulting in hypoperfusion and hypoxia, promote new dermal collagen formation and rapid differentiation of keratinocytes
Losartan [181]	Fibroblasts with decreased contractile activity, migration, and adhesion
Oxandolone + hyaluronic acid gel [182]	Decrease inflammation, collagen and fibroblast cellularity, vascularization, and myofibroblast activity
Dipeptidyl peptidase-4 inhibitors [183]	Attenuate collagen synthesis and deposition
1,4-Diaminobutane [184]	Inhibits collagen cross-linking
Autologous fat grafting [185]	Decreases the expression of the myofibroblast marker α-SMA * and ECM * components
Stem cells [186,187]	Reduce inflammation, cellularity, and collagen filaments
Platelet-rich plasma [61,188,189]	Reduces expression of TGF-β1 * and CTGF mRNA
CD206 * + Macrophages and Fibroblasts [190]	Increase MMP-1 * and decrease expression of pro-fibrotic factors, COL1A1 *, COL2A1 *, α-SMA *, CTGF *, and TGF-β1 *
IL-6 * [190]	Increases expression of anti-fibrotic genes
IFN-γ * [191]	Increases expression of COL1A1 * and COL3A1 * mRNA and decreases expression of MMP-2 * (gelatinase)

* Abbreviations: CO_2_ (carbon dioxide); CD206 (cluster of differentiation 206); IL-6 (interleukin 6); IFN-γ (Interferon gamma); α-SMA (alpha smooth muscle actin); ECM (extracellular matrix); TGF-β1 (transforming growth factor beta-1); COL1A1 (collagen type I alpha 1 chain); COL2A1 (collagen type II alpha 1 chain); COL3A1 (collagen type III alpha 1 chain), CTGF (connective tissue growth factor); TGF-β1 (transforming growth factor beta 1); MMP-2 (matrix metalloproteinase 2).

## Data Availability

Not applicable.

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
