# Peer review of "An Updated Review of Hypertrophic Scarring"

_cells, 2023, doi:10.3390/cells12050678_

Round 1

Reviewer 1 Report

This review article aims to provide an updated review of  the literature pertaining to the underlying mechanism hyperthrofic scaring of hyperthrofic scaring, animal models of hyperthrofic scaring, and the  current and emerging treatments of hyperthrofic scaring

The work is well written and reads very well. The English adopted by the authors is clear and comprehensive and therefore few changes will be necessary for any publication. The topic itself is not known and consequently the authors are able to give a very clear general picture.

As an expert, I greatly appreciated the description of wound healing triggered in the cellular mechanisms related to the formation of hypertrophic scars. The addition of a chapter dedicated to therapies completes the ambitious manuscript. The iconography is done very well and consequently there is no need to clarify further.

The only suggestions are those related to bibliographic updating as many of the items reported are obsolete and to dedicate some reflection on the role of mast cells in the dynamics of cellular mechanisms related to wound healing.

.

I have two questions:

Authors should check reference entries. Many of them are outdated. At least the 50% of references should be less than 5 years old

What is the role of mast cells during the processes related to hyperthrofic scar?

Author Response

Response: We thank the reviewer for the thorough review of our manuscript and for his/her insightful comments. Addressing these comments/queries has improved our manuscript and we hope the reviewer agrees. Below, please find our point-by-point responses to your queries.

  • Authors should check reference entries. Many of them are outdated. At least the 50% of references should be less than 5 years old.

Response: We appreciate the reviewer’s comment in this regard but respectfully disagree that these references are “outdated”. We included some older references because they have fundamentally contributed to our current understanding of hypertrophic scar. We have also discussed the more recent findings so as to update the current knowledge that was build upon the previous knowledge. We believe that including the older references allows the readers to evaluate the original research articles if they so choose, but it will also highlight the need for follow-up research in the field.

  • What is the role of mast cells during the processes related to hypertrophic scar?

Response: There is some controversy regarding the role of mast cells in hypertrophic scar development. Most published studies in this area support a pro-fibrotic role for mast cells in the skin, as many mast cell-derived mediators stimulate fibroblast activity and studies generally indicate higher numbers of mast cells and/or mast cell activation in scars and fibrotic skin. However, some studies in mast cell-deficient mice have suggested that these cells may not play a critical role in cutaneous scarring/fibrosis. In lieu of these controversies and since there is a recent review article on the evidence for and against a role for mast cells in cutaneous scarring and fibrosis [1], we did not think it was appropriate for us to repeat these points and thus we opted not to include these discussions in this review article.

  1. Wilgus, T.A., S. Ud-Din, and A. Bayat, A review of the evidence for and against a role for mast cells in cutaneous scarring and fibrosis. International Journal of Molecular Sciences, 2020. 21(24): p. 9673.

Reviewer 2 Report

I have reviewed an interesting review by Manjula P. Mony et al. entitled “An Updated Review of Hypertrophic Scar”. Here are my comments:

-       Author panel and affiliations should be corrected.

-       Line 15: “since the last review of hypertrophic scar…” - Whose last review?

-       Abstract section is extremely vague and should be rewritten in order to be more attractive to readers.

-       The second paragraph of the introduction (lines 37-54) section should be split for a better understanding.  

-       The objectives of this review should be divided into primary and secondary because at the moment it appears as a string of ideas.

-       Lines 58 - “To better our understanding” – please correct.

-       Line 59 – not sure about the need of saying “we provide an overview…”; the name of this subsection allows us to understand that this is what it's about.

-       Line 61 and more – please try to avoid “;” in the middle of a paragraph, and to use full sentences. This is a scientific paper.

-       Did the figures have a source of inspiration, or are they the authors' own creation?

-       Lines 89-90 – “within minutes, but peaks within”- try to rewrite in order to avoid repeating words

-       Lines 70-87 – maybe this paragraph can be split into primary and secondary hemostasis.

-       Lines 89-111 – it is hard to follow the information in this paragraph. Maybe the early and the late phase of the inflammation can be addressed in two separate ideas.

-       Lines 140-143 – try to rewrite this part, especially “which are themselves tightly regulated by the tissue….”

-       Lines 145-151 – “over time…..”, “concurrently and over time…” – please rewrite

-       Lines 153-155 – why repeating the same information from the introduction section?

-       Lines 155-158 – these ideas are very confusing and not very related one to another. Please rewrite and move them to the introduction section.

-       Lines 152-178 – I think that this entire section would be more fitted into the introduction section, in order to emphasize the importance of this review.

-       Lines 180-183 – please find a synonym for the word “excessive”.

-       Line 207 – explain TLR

-       Lines 210-214 – explain the abbreviations not previously encountered in the text.

-       Lines 237-251 - explain the abbreviations not previously encountered in the text.

-       Lines 251-252 – please provide more information about fatty acid-derived inflammation, and it’s role in influencing scarring formation. This is an updated review. It is not enough only to cite a paper. More data about on this topic are required, specially about the expression of SREBP1 or FASN.

-       Lines 254-259 – try not rewrite (and to find a synonym for “during”)

-       Line 286 – explain MMP

-       Lines 290-291 – “During HTS… During this phase,….” – please rewrite

-       Lines 297-308 - explain the abbreviations not previously encountered in the text.

-       Every figure should have a footer and all the abbreviations with the figure should be explained here (Figure 3)

-       Line 322 – “In past several decades” – please rewrite

-       Lines 322-348 – maybe this can be split: information about rabbit, pig and rodents models.

-       Very table should have a footer and all the abbreviations should be explained within it (Table 1).

-       Line 401 – “pre-existing”

-       Lines 401-402 – “Scar revision is the simplest method of treating preexisting HTS and encompasses procedures aimed at excisional debulking of hypertrophic scar tissue ([165] and Table 2).” - Can't corrective surgery lead to new hypertrophic scarring? Some comments on this issue are necessary.

-       Line 412 – explain CO2

-       Lines 414-415 – not sure about this statement “The use of pharmacologic agents in the treatment of hypertrophic scars has also been well-established”. If the role was so well established, we might be talking about a lower incidence of hypertrophic scarring, no?

-       Line 421 – “a recent literature review and meta-analysis found that ” – a review can emphasize something already “found” before – please rewrite

-       Table 2 – explain the abbreviation

-       line 504 – explain BMP and

-       Table 3 – explain abbreviation in a footer under the table.

-       Conclusion section – please point out the main development about the hypertrophic scarring from the last years.

-       References should be described according to publisher’s indications (https://www.mdpi.com/journal/cells/instructions)

-       “Acknowledgments: We would like to acknowledge the help that Okensama La-Anyane and Ayaan Ahmed provided in putting together this manuscript.” – what do you mean by putting together the manuscript?

Author Response

We thank the reviewer for the thorough review of our manuscript and for his/her insightful comments. Addressing these queries have improved our manuscript as we hope the reviewer agrees. Below, please find our point-by-point responses to your queries.

Author panel and affiliations should be corrected.

Response: We have revised the author panel and affiliations according to the Cells’ formatting guidelines.

-       Line 15: “since the last review of hypertrophic scar…” - Whose last review?

Response: We apologize to the reviewer for this oversight. We have revised the introduction section of the revised manuscript to address the reviewer’s query.

-       Abstract section is extremely vague and should be rewritten in order to be more attractive to readers.

Response: As the reviewer appreciates, the abstracts of the review articles cannot be too specific, as they provide a general overviews of specific topics. With that said, we agree with the reviewer that the abstract in the original manuscript was rather vague and not that informative and have revised the abstract to comply with the reviewer’s suggestion.

-       The second paragraph of the introduction (lines 37-54) section should be split for a better understanding.

Response: Per reviewer’s suggestion, we have split this section into 2 parts.

-       The objectives of this review should be divided into primary and secondary because at the moment it appears as a string of ideas.

Response: We have added a paragraph at the end of the introduction section, delineating the sequence of review as they relate to HTS.

-       Lines 58 - “To better our understanding” – please correct.

Response: We have replaced this to read “To gain better understanding of…”

-       Line 59 – not sure about the need of saying “we provide an overview…”; the name of this subsection allows us to understand that this is what it's about.

Response: Per reviewer’s suggestion, we have revised this sentence.

-       Line 61 and more – please try to avoid “;” in the middle of a paragraph, and to use full sentences. This is a scientific paper.

Response: We have revised as suggested by the reviewer.

-       Did the figures have a source of inspiration, or are they the authors' own creation?

Response: These are original figures which we have generated for this review article and have never been published before.

-       Lines 89-90 – “within minutes, but peaks within”- try to rewrite in order to avoid repeating words

Response: We have revised as suggested by the reviewer.

-       Lines 70-87 – maybe this paragraph can be split into primary and secondary hemostasis.

Response: We have revised as suggested by the reviewer.

-       Lines 89-111 – it is hard to follow the information in this paragraph. Maybe the early and the late phase of the inflammation can be addressed in two separate ideas.

Response: We have revised this section.

-       Lines 140-143 – try to rewrite this part, especially “which are themselves tightly regulated by the tissue….”

Response: We have revised this section.

-       Lines 145-151 – “over time…..”, “concurrently and over time…” – please rewrite

Response: We have revised as suggested by the reviewer.

-       Lines 153-155 – why repeating the same information from the introduction section?

Response: We have removed the repeating information in the revised manuscript.

-       Lines 155-158 – these ideas are very confusing and not very related one to another. Please rewrite and move them to the introduction section.

Response: This section has been removed and is now included in the introduction page.

-       Lines 152-178 – I think that this entire section would be more fitted into the introduction section, in order to emphasize the importance of this review.

Response: This section has been removed and is now included in the introduction page.

-       Lines 180-183 – please find a synonym for the word “excessive”.

Response: We respectfully believe that “excessive” is the proper word to use in this sentence.

-       Line 207 – explain TLR

Response: Revised to read Toll Like Receptor (TLR)

-       Lines 210-214 – explain the abbreviations not previously encountered in the text.

Response: Revised as suggested by the reviewer.

-       Lines 237-251 - explain the abbreviations not previously encountered in the text.

Response: Revised as suggested by the reviewer.

-       Lines 251-252 – please provide more information about fatty acid-derived inflammation, and it’s role in influencing scarring formation. This is an updated review. It is not enough only to cite a paper. More data about on this topic are required, specially about the expression of SREBP1 or FASN.

Response: We have added the following to the revised manuscript. “Moreover, alteration in the fatty acid metabolism influences inflammation and can result in excessive scarring [98, 99]. In a recent study, the expression of sterol regulatory element-binding protein-1 (SREBP1) and fatty acid synthase (FASN) was shown to be reduced at mRNA and protein levels in pathological HTS and in HTS-derived fibroblasts [98]. In another study, the expression of fatty acid desaturase 1 and 2 (FAD1 and FAD2) - (key enzymes in the polyunsaturated fatty acids (PUFAs) metabolism with demonstrated anti-inflammatory function [100]) - were lower in keloids and keloid-derived fibroblasts [99].”

-       Lines 254-259 – try not rewrite (and to find a synonym for “during”)

Response: We have revised it as suggested by the reviewer.

-       Line 286 – explain MMP

Response: matrix metalloproteinases (MMPs). Please see Line 168.

-       Lines 290-291 – “During HTS… During this phase,….” – please rewrite

Response: We have revised it as suggested by the reviewer.

-       Lines 297-308 - explain the abbreviations not previously encountered in the text.

Response: We have revised it as suggested by the reviewer.

-       Every figure should have a footer and all the abbreviations with the figure should be explained here (Figure 3)

Response: We have included figure legends for all the figures but we did not discuss the players in the legends because we wanted to avoid redundant statements as they were all discussed in the text. In addition, all abbreviations were fully expanded in the text upon their first use.   

-       Line 322 – “In past several decades” – please rewrite

Response: We have revised it to read “In the past several decades”

-       Lines 322-348 – maybe this can be split: information about rabbit, pig and rodents models.

Response: We have revised this section.

-       Very table should have a footer and all the abbreviations should be explained within it (Table 1).

Response: We have included a footer for all the Tables to expand the abbreviations that were included in the Tables.

-       Line 401 – “pre-existing”

Response: We did not find “pre-existing” in the text however we assumed that the reviewer wishes us to change “preexisting” to “pre-existing” in this sentence. We made this correction.

-       Lines 401-402 – “Scar revision is the simplest method of treating preexisting HTS and encompasses procedures aimed at excisional debulking of hypertrophic scar tissue ([165] and Table 2).” - Can't corrective surgery lead to new hypertrophic scarring? Some comments on this issue are necessary.

Response: To be effective, scar revisions should be performed over 1 year from the original injury to give adequate time for the scar to mature [163], as immature scars are prone to hypertrophic healing and give poor results after scar revision [171].

-       Line 412 – explain CO2

Response: Carbon dioxide (CO2).

-       Lines 414-415 – not sure about this statement “The use of pharmacologic agents in the treatment of hypertrophic scars has also been well-established”. If the role was so well established, we might be talking about a lower incidence of hypertrophic scarring, no?

Response: We have revised the sentence to read; “Pharmacologic agents have also been used frequently in the treatment of hypertrophic scars, with common agents including corticosteroids, chemotherapeutic agents, and Botulinum toxin-A.”

-       Line 421 – “a recent literature review and meta-analysis found that ” – a review can emphasize something already “found” before – please rewrite

Response: Revised as suggested.

-       Table 2 – explain the abbreviation

Response: Abbreviations were added in a footer under the table.

-       line 504 – explain BMP and

Response: BMP-4 (Bone morphogenetic protein 4). We have revised the text accordingly.

-       Table 3 – explain abbreviation in a footer under the table.

Response: Abbreviations were added in a footer under the table.

-       Conclusion section – please point out the main development about the hypertrophic scarring from the last years.

Response: Revised as suggested.

-       References should be described according to publisher’s indications (https://www.mdpi.com/journal/cells/instructions)

Response: Revised as suggested.

-       “Acknowledgments: We would like to acknowledge the help that Okensama La-Anyane and Ayaan Ahmed provided in putting together this manuscript.” – what do you mean by putting together the manuscript?

Response: The acknowledgment has been modified in the revised manuscript.
